# Viroids and the Origin of Life

**DOI:** 10.3390/ijms22073476

**Published:** 2021-03-28

**Authors:** Karin Moelling, Felix Broecker

**Affiliations:** 1Institute of Medical Microbiology, University of Zurich, Gloriastr 30, 8006 Zurich, Switzerland; 2Max Planck Institute for molecular Genetics, Ihnestr. 63-73, 14195 Berlin, Germany; 3Vaxxilon Deutschland GmbH, Magnusstr. 11, 12489 Berlin, Germany; felixbroecker@gmx.net

**Keywords:** ribozymes, RNase P, early earth, exoplanets, endosymbionts, non-coding RNA, meteorites

## Abstract

Viroids are non-coding circular RNA molecules with rod-like or branched structures. They are often ribozymes, characterized by catalytic RNA. They can perform many basic functions of life and may have played a role in evolution since the beginning of life on Earth. They can cleave, join, replicate, and undergo Darwinian evolution. Furthermore, ribozymes are the essential elements for protein synthesis of cellular organisms as parts of ribosomes. Thus, they must have preceded DNA and proteins during evolution. Here, we discuss the current evidence for viroids or viroid-like RNAs as a likely origin of life on Earth. As such, they may also be considered as models for life on other planets or moons in the solar system as well as on exoplanets.

## 1. Introduction

Viroids, which are known today as virus-like infectious agents of mainly plants, may be considered as remnants of the Ancient RNA World that is thought to have existed before the emergence of DNA and proteins [1]. Naturally occurring viroids can be enzymatically active, i.e., act as ribozymes, and catalyze the cleavage of RNA [2]. In addition, it has been shown experimentally that in vitro generated RNAs can perform various additional enzymatic activities, including RNA cleavage and ligation, the formation of peptide bonds, even polymerize their own copies, and many more [3,4]. The RNA of viroids and ribozymes is typically non-coding; information is based on sequence and structure. In theory, any chemical reaction within biotic agents can be carried out by ribozymes. According to the Ancient RNA World hypothesis, RNA has been the first biomacromolecule on Earth, appearing during pre-cellular evolution of life [1]. It is likely that the building blocks of complex biopolymers such as RNA have originated from specific regions in the ocean of early Earth, such as hydrothermal vents that provide an energy source (i.e., heat) as well as the necessary chemical precursors, elements, and metal ions required for catalysis. Local niches may have acted as Darwin’s “warm little ponds” and increased the concentration of reagents. In a water-ice mixture that a ribozyme was selected for, which was able to polymerize a product as long as itself, functioning as an RNA replicase, another important step for the origin of life [5]. The formation of organic molecules, including precursors of RNAs, has been shown under such simulated conditions by a number of experiments including the classical Miller–Urey experiment which achieved the synthesis of amino acids under laboratory conditions [6,7]. Nucleobases were later generated in the laboratory, for example, by Sutherland, who performed a “one pot” synthesis leading to amino acids, nucleosides, and lipids [8]. Alternatively, or additionally, the chemical analysis of meteorites suggests that some of the small molecule building blocks for RNA and other biopolymers may have originated from outer space [9], whereas complex RNA structures and other biopolymers are unlikely to survive travel through space. Exposure to cosmic rays would destroy the RNA. Thus, while some of the small molecular buildings blocks for life, including amino acids, may have come to early Earth via meteorites, complex biopolymers must have evolved on Earth. Ribozymes likely constitute one of the first types of biomolecules that have evolved, as evidenced by the fact that they do not have coding capacity and that protein synthesis in all known cellular life on Earth still relies on ribozymes, the catalytically active part of the ribosome. Another ribozyme processes the pre-transfer RNA in all cellular organisms.

There is evidence from sequence analyses that viroid-like RNAs have evolved into viruses [10] and that during evolution viruses or their ancestors predate the appearance of the last universal cellular ancestor (LUCA) [11]. These early virus-like entities have likely evolved protein and DNA synthesis in the absence of cells. The discovery of giant viruses that have many features previously thought to be restricted to cells, such as expression of components of the protein synthesis machinery and metabolic molecules, has further supported the idea that the evolution from viruses to cells is continuous [12,13,14]. Viral evolution likely occurred before the evolution of cells [11]. There is an alternative hypothesis about the existence of virocells, which states that viruses evolved after ribosome-encoding cells [15]. If, however, viruses or virus-like entities arose first during evolution, with cells being a later evolutionary achievement, the origin of life may be something similar to the simplest virus-like entities we know. These are the viroids or ribozymes, which are composed of RNA and typically do not encode any proteins. It is therefore likely that catalytically active viroid-like structures, i.e., the ancestors of present-day viroids that have been suggested to have existed before the evolution of cells [16]. They may have played a role at the origin of life on Earth and during evolution. That is why Diener described the viroids as “living fossils” from early stages of evolution [16]. Consequently, we suggest that the search for extraterrestrial life should include the detection of viroid-like entities or analogous elements.

Viroids were first recognized as disease-causing agent in potatoes as potato spindle tuber viroid (PSTVd) [17]. Ribozymes and viroids share many structural properties. Both are circular, non-coding RNA molecules whereby the viroids are larger, ranging from 246 to 434 nucleotides, all containing a conserved central region CCR. They are larger than the ribozymes, which consist of a core and three helices, in total 50 to 150 nucleotides.

The viroids consist of a self-complementary rod-like or branched structured covalently closed circular RNA. They comprise two families, but only one of them harbors an enzymatically active hammerhead motif, the other one relies on host factors. Their names refer to their hosts, avocados or potatoes (*Avsun-* or *Pospiviroidae*, respectively). The catalytic activity of the hammerhead structure is a common structural motif of the catalytically active viroids or ribozymes. The catalytic activity of the hammerhead in the *Avsunviroides* such as Avocado Sun Blotch Viroid (ASBVd) is involved in its replication and can lead to a self-cleavage mechanism to monomeric or dimeric forms, together with its own RNA ligase. A member of the other group is the Potato Spindle Tuber Viroid (PSTV), which is characterized by a conserved central region CCR without enzymatic activity or a hammerhead motif [18]. Thus, a whole group of viroids has no enzymatic activity. Viroids rely on host enzymes for replication through a “rolling circle” mechanism by RNA polymerase II, which normally copies DNA, not RNA, as in this case from the viroid RNA [19]. Environmental factors may have allowed for loss of enzyme functions. They may get lost by functional regression and adaptation to the presence of host enzymes [2,20].

The hammerhead motif has been shown to be a universal motif in the tree of life including its presence in bacteria and animals such as corals, nematodes, and arthropods, as well as in plant genomes [21]. The enzymatic hammerhead motif is also typical of most ribozymes, only some of them are characterized by a special hairpin-type motif. Related hammerhead sequences have also been found in mobile genetic elements in mammalian genomes such as the short interspersed nuclear elements (SINEs) [21]. This widespread distribution among the tree of life further supports an ancient evolutionary origin of hammerhead ribozymes. Today, the viroids are mainly known as pathogens in plants, while the ribozymes are smaller and consist of a core with three double-stranded helices. They are not known as pathogens. Both of them can cleave themselves autocatalytically [2,22].

Ribozymes are the catalytically active component of ribosomes, the protein synthesis complex, making them one of the most important factors for evolution of protein synthesis before the existence of cells [23].

Today’s viroids act as plant pathogens and cause diseases presumably by leading to the formation of viroid-derived small RNAs, which play a role in pathogenicity mediated by small interfering siRNAs, which cause silencing of important host genes involving RISC, the RNA-Induced Silencing Complex. It mediates transcriptional or post-transcriptional dysregulation of host protein expression, although other yet unknown mechanisms cannot be excluded [24,25].

The viroids themselves are very resistant to RNA silencing-mediated degradation. The self-complementary rod-like structure of the covalently closed circular RNA is a robust secondary structure which had a selective advantage to survive during evolution.

## 2. The Ancient RNA World

Most of the theories on the origin of life center around hydrothermal vents or “black smokers”, the volcanoes deep under the sea level [26,27]. They offer strong temperature gradients, high pressures, elements that act as catalysts, and diverse building blocks of life.

At this point it may be worth defining life, for which there are dozens of definitions. The one by the National Aeronautics and Space Agency (NASA) is: “Life is a self-sustaining chemical system capable of Darwinian evolution”. Two origins of life were proposed by theoretical physicist Freeman Dyson in cooperation with the mathematician John von Neumann. They were creating this concept together in Princeton [28]. These two origins of life are based on the concept of computers: Machines as hardware and programs as software. RNA is the only biological molecule we know of that is both “software and hardware” in one. RNA can be a self-replicating chemical entity, and at the same time also a carrier of information.

The Nobel Prize laureate Jack Szostak asked the question what kind of structures can be formed by RNA in vitro. He performed an experiment with stretches of RNA in a reaction vessel, starting with a large pool of 10^15^ random RNA molecules, 220 nucleotides in length [3]. The RNA molecules can adopt all kinds of conformations. The pool of RNAs represents a quasispecies of different theoretically possibly sequences, a huge sequence space exceeding all information on Earth [29,30]. By in vitro selection for functional nucleic acids forming under laboratory conditions some RNA molecules were shown to be catalytically active RNA enzymes, the ribozymes. One in 10^10^ random RNA molecules adopted a conformation that formed a binding site for a randomly chosen set of substrates [3]. The ribozymes can catalyze a diversity of reactions, including the cleavage and ligation of nucleic acids, RNA synthesis, and peptide bond formation. Thus, ribozymes can direct a primitive metabolism before the evolution of protein and DNA synthesis. Other RNA configurations may have resulted in structures such as transfer RNAs (tRNAs), which are processed from a pre-tRNA by enzymatic RNA, the RNase P, which is also a ribozyme [3,31].

“RNA can do it all” is a sentence attributed to Sir Francis Crick, one of the discoverers of the DNA double-helix structure [32]. The smallest replicating entities one can think of are the viroids or ribozymes, both of which can exhibit catalytic enzymatic functions. Viroids or ribozymes are composed of naked closed circular RNA with zero genes, i.e., they do not code for triplets for amino acids as we know them on Earth. They not contain genetic, i.e., coding, but structural information [14].

The RNAs exhibit structure-mediated functions. Ribozymes were tested by Gerald F. Joyce in a test tube, where they proved again to be multifunctional; they can cleave, join, chemically replicate other ribozymes, thus generating progeny, form peptide bonds, and mutate depending on environmental conditions [33,34]. It has been suggested that ribozymes can even replace the function of a replicase and perform RNA polymerization. Additionally, DNA has been proposed to have arisen before a reverse transcriptase enzyme existed by chemical deoxygenation of ribonucleotides [3]. Similar functions of ribozymes have recently been attributed to desoxyribozymes, “DNAzymes”, which can cleave and join and show similar properties as RNA ribozymes [35].

The question on energy sources is not answered, it is not even mentioned by the NASA definition of life above, yet any replication process would require an energy supply. The energy during early evolution was unlikely provided by the Sun, as sunlight can only reach to about 200 m below sea level. Life, however, likely started at hydrothermal vents deep in the ocean where sunlight cannot reach, where, for example, geothermal energy is available. In addition, ice-cold water allows cold-adapted ribozymes to catalyze rather long RNA in vitro [36].

## 3. Meteorites

Some of the molecules which may have existed on the Early Earth can be identified by analyzing meteorites. A huge meteorite (over 100 kg) spread fragments over a large area in South-West Australia in 1969, the Murchison meteorite. Chemical analyses by various research groups indicated the presence of more than 80 amino acids—a surprising number considering the 20 amino acids used by biological entities (cellular organisms and viruses) on Earth [37,38,39,40]. The Murchison meteorite also contained purines and pyrimidines, the building blocks for nucleic acids, sugar-related organic compounds, and there were di-amino acids, aromatic hydrocarbons, and up to 14,000 unique carbon-containing compounds [9]. A large one was a structure consisting of hexameric and pentameric carbon rings, an icosahedron with a total of 60 carbons, designated as fullerene [41]. This structure is known from soccer balls, but also as energetically optimal structure of the capsid of some viruses, such as the retroviruses. Indeed, there are scientists who seriously suggested that during the Cambrian Explosion of life around 500 Million years ago, retroviruses came from the outer space as “rain” falling onto the Earth. This example of the “Panspermia hypothesis” would not answer the question how life started on Earth, but would simply shift the problem away from Earth to another celestial body. During their passage through space, the viruses would be protected inside meteorites against cosmic rays, ionizing radiation or other destructive influences—that is the hypothesis [42]. The stability of retroviruses or nucleic acids, however, is unlikely to be strong enough to survive such a passage through space [43].

The question of whether the building blocks that are detectable on meteorites are sufficient for the formation of life on Earth as we know it prompted the group of John D. Sutherland and colleagues in the United Kingdom to perform a synthesis of the major biomolecules for life on Earth: Amino acids, lipids, and ribonucleotides, the components for proteins, fat, and nucleic acids, respectively. The necessary elements, carbon, hydrogen, oxygen, phosphorous, nitrogen, and sulfur (*C*, *H*, *O*, *P*, *N*, *S*), were present as simple precursors such as hydrogen cyanide, inorganic phosphate, and hydrogen sulfide in a so-called “one pot synthesis” vessel and indeed allowed for the synthesis of the necessary biomolecules for living matter, RNA, proteins, and lipids [8]. These are likely to exist on many other stars or exoplanets with an elemental composition similar to that of Earth, based on the predicted abundance of the different elements throughout the universe. Especially the detection of exoplanets stimulated the question about the existence of extraterrestrial life [44]. The Big Bang and our universe started simple: With hydrogen containing one proton with one electron, and the few elements mentioned above that are derived from it. Some of the bio-elements necessary for life are multiples of hydrogen, resulting in helium, nitrogen, oxygen, and carbon.

More components are not necessary as minimal building blocks of life. The periodic table with all the elements found on Earth is valid throughout the whole universe—except that the developmental stages of other solar systems vary depending on the history and evolution of the stars, and not all of them exhibit similarities to our Sun.

The one pot-synthesis experiment mentioned above had a precedent, the Miller–Urey experiment from 1959. It achieved the synthesis of organic compounds in the laboratory from simple inorganic precursors, such as water (H_2_O), methane (CH_4_), ammonia (NH_3_), and hydrogen (H_2_) under putative early Earth-like conditions, precursors which may have dominated the atmosphere of the early Earth. Artificial lightning served as a source of energy for chemical reactions. Among the many biomolecules, five different amino acids were generated [6,7].

## 4. Ribosomes Are Ribozymes

One can speculate that an amino acid became attached to an RNA, possibly a transfer RNA- (tRNA)-like molecule that supported the transfer of the amino acid to a ribozyme, which fixed a peptide bond to another amino acid. This would be a primitive “pre-ribosome”. It is surprising that the protein synthesis machinery, the ribosomes as we know them today, use a catalytic RNA, a ribozyme, for peptide bond formation, even though RNA enzymes are typically less efficient than protein enzymes. Protein enzymes do not exist in the ribosomes. “Ribosomes are ribozymes” is the summary by Thomas Cech making them one of the most important factors for evolution and protein synthesis and precedents of cells [23].

Ribozymes were found in the introns of RNA transcripts and can remove themselves, and are involved in the maturation of pre-tRNAs as component of the RNase P complex. They can catalyze their own synthesis as RNA polymerase ribozyme and they cleave themselves autocatalytically [2,22].

In today’s cells, the tRNA is processed from a pre-tRNA precursor by a ribozyme, the RNase P. It was Sidney Altman who discovered this ribozyme and shared the Nobel Prize in Chemistry in 1989 with Thomas Cech “for their discovery of catalytic properties of RNA”. Until then it was thought that all enzymes were proteins. The RNase P is closely associated with a protein, which however is enzymatically inactive. The RNA moiety alone harbors the catalytic activity.

The first substrate of the RNase P was discovered by Altman, and he defined it as a very ancient structure: A tRNA precursor, pre-tRNA. He showed that the RNase P recognizes structures of this substrate and cleaves at a conserved site next to a loop of the molecule. Only later evolutionarily, in the protein world, was the RNase P associated with a protein. This is reminiscent of the catalytic ribozyme in the ribosomes—which finally accumulated about one hundred proteins as scaffold molecules [45].

It is often discussed that RNA enzymes (ribozymes) evolved into proteins, which performed similar functions, but much more efficiently. For the RNase P, it was directly shown at the molecular level how the ribozyme evolved into a protein. Comparison of the role of metal ions and properties of the scissile bond indicated similarities between the ribozyme-based and a protein-based RNase. This indeed supports the notion that RNA can be a precursor of proteins. The protein’s catalytic activity is highly increased as demonstrated by Koutmos et al. [46,47]. Thus, two ribozymes have played important roles in the evolution of protein synthesis and the ribosomes. This supports their importance in the pre-protein world.

Other non-coding ribosomal RNAs exist, and about one hundred proteins serve some functions for structure and stability, but not for catalysis, resulting in the composition of today’s ribosomes. Thus, ribozymes, which originate from the Ancient RNA World, are even in the DNA and protein world that later evolved the essential components to produce proteins. Ribozymes must have been first (Figure 1).

DNA could also have been produced without proteins by chemical deoxygenation of RNA. Thus, the transition from the RNA to the DNA and protein world may have taken place before the protein enzyme evolved, which catalyzes this conversion, the reverse transcriptase (RT) [3]. The reverse transcription of RNA into DNA is the essential step in retroviral replication from RNA to DNA and coined the name of the virus, the retroviruses. Yet, the RT is of much more general importance for the mobility of retrotransposons, the elongation of telomeres, and others [48]. Protein enzymes accelerated reactions compared to RNA enzymes, and thereby were able to speed up evolution.

We happened to observe the effect of a protein on a ribozyme when trying to evaluate a ribozyme for gene therapy [49]. We supplemented a ribozyme with a protein based on the fact that all RNA viruses always carry proteins in the RNA. RNA-binding proteins with a net positive charge are commonly found inside viral cores, stabilizing the viral RNA and serving as enhancer of replication. Such an RNA binding protein was developed as stimulator of a hammerhead ribozyme used for gene therapy, which works by cleaving an mRNA involved in chronic myeloid leukemia. One such protein is the nucleocapsid NCP7 of human immunodeficiency virus, a protein which activated the catalytic activity of the ribozyme by a hundredfold or more. RNA-binding proteins are often positively charged, rich in R (arginine) and K (lysine) to bind well to the negatively charged RNAs. Some of them are zinc finger-type proteins [49]. They are the typical nucleocapsid proteins lining the viral RNAs inside the virions, protecting them against nucleases. They also have a “matchmaker” or chaperone function by disentangling RNA hyper-structures and allowing RNA transcription with much higher efficiency, as was shown for reverse transcription by the RT of viral RNA [49]. Such positively charged amino acids may have accelerated the catalytic activity of ribozymes on early Earth.

A few molecules are known, which appear to be a chimera of a protein with RNA, perhaps “left-overs” from the time of the transition of the RNA world to the RNA-protein world, as pointed out by Szostak in a university lecture, vitamin B12 and acetyl-CoA. They have RNA tails, as if removal of the RNA moiety had been incomplete, and now may serve a function such as promoting stability [50].

It is worth mentioning that the first identified ribozyme was that of the satellite RNA of tobacco ringspot virus, consisting of a 359 nucleotide single-stranded covalently closed circular genome that is parasitic upon the tobacco ringspot virus [51]. The catalytic satellite RNA genomes cleave themselves. Satellite RNAs are plant parasites which depend on an associated helper virus [52].

There are still today other plant viruses with structures similar to ribozymes, the viroids, and others such as Narnaviruses or viruses with tRNA-like-structures [53,54], which may be remnants of the Ancient RNA World. There are viral non-coding sequences in virtually all RNA viruses, especially at their termini where, for example, regulatory factors can bind [55]. However, even next to coding regions, non-coding regions can undergo evolution to become coding for a protein or peptide. Thus, this kind of evolution from non-coding to amino acid-coding RNA is still ongoing today. Evolution never ends and can be witnessed in the rapidly evolving viruses. Viroids play a role in many biological processes; most noticeably as the cause of diseases of plants. The pathogenicity domain next to the conserved center of viroids plays a role in diseases, mediated partly by siRNAs, which causes silencing of important host genes involving RISC, the RNA-induced silencing complex [24,25].

Known from history is Cadang-Cadang, a disease of trees caused by a viroid in the Philippines affecting coconut trees, where the viroid was transmitted by the workers during harvesting, possibly by using tools such as a knife to cut off the fruits of many trees, which might have become cross-contaminated [56,57]. Transmission of the viroids destroyed whole plantations. The disease is not only of historical interest, but still ongoing in Indonesia, Malaysia, West Africa, Central America, and further regions. Many other viroids are known, infecting various plant species.

An interesting case is the effect of a viroid together with another virus on the patterns of carnation [48]. Carnation flowers can be infected by a viroid-related RNA, the Carnation small viroid-like (CarSV) RNA, 275 nucleotides long, which can occur also in DNA form. It is thus a retroviroid-like element, a unique structure, and the DNA form is probably provided by an RT of a plant pararetrovirus, the caulimovirus, for reverse transcription of the viroid RNA to DNA [58]. Whether loss of the red color around the rims of the flowers of the carnations might be influenced by some viroid-mediated silencing mechanism remains to be analyzed (Figure 2).

Alternatively, novel functions of viroids may be acquired by uptake of coding information from the host, as has likely been the case for the hepatitis delta virus (HDV). The HDV genome shares similarities with viroids and also has ribozyme activity [59]. It has thus been suggested that it likely originated from a catalytically active plant viroid and acquired coding information from a host RNA [60]. Although a viroid as origin appears most likely, other hypotheses suggest that HDV may have originated from other RNAs, such as a virusoid or a host mRNA [61]. The HDV gene expresses proteins supporting processing and transport and allow the viroid to perform novel functions. Normally, viroids comprise 250 to 400 nucleotides, yet HDV comprises about 1700 nucleotides [2]. Co-infection with the hepatitis B virus in the liver allowed the ribozyme to adopt a coat and to become mobile, i.e., move between cells and organisms [62]. Thus, if HDV originates from a viroid, it appears possible that viroids can take up cellular genes and increase in size as evidenced here. Coding and non-coding RNA regions are characteristic of almost all RNA viruses, with non-coding RNA at the 3′- and 5′-ends being responsible for regulatory information and for stability of the RNA genomes. Non-coding RNA and DNA can even today evolve to coding information [63,64], thus proving that evolution is always going on (Figure 2).

Structures of non-coding RNAs are abundant in all biological systems. They are reminiscent of viroid RNAs. They can act as regulators of gene expressions, such as siRNAs, or function as guardians of gene expression as described for circular RNAs (circRNAs) [65]. These circRNAs have recently emerged as a large class of animal RNAs with complex tissue- and stage-specific expression patterns. One human circRNA has recently been identified as “sponge”, as a negative regulator of microRNAs such as miR-7. Many binding sites, up to 70 for miR-7 in one circRNA molecule, makes its binding and regulator effect very efficient. Many other circRNAs have been identified in plants as well, although it is unclear whether acting as miRNA sponge is their main function [66]. It is presently unknown whether viroids can also act as miRNA sponges, but due to their structural similarities with circRNAs, this possibility cannot be excluded. Already in 1976, Sänger discovered covalently closed circular RNAs in plant viroids; such circRNAs were later found in all kingdoms of life [67,68] (Figure 2).

## 5. Dilemma of “Chicken-or-Egg First”?

There is a chicken-and-egg dilemma, since present viruses are parasites that require cells to replicate. How then can viruses have been first [69]? The ancestors of present-day viruses may have been free-living entities originally which later during evolution lost their autonomy and became intracellular parasites after the evolution of cells. In support of this theory, there is evidence that the ancestors of present-day viruses have shown extensive evolution, including from RNA to DNA virus-like entities, before the emergence of the last universal cellular ancestor (LUCA) [11]. Likewise, the ancestors of viroids have been suggested to be evolutionarily older than cellular organisms [16].

If one assumes the ancestors of viruses and viroids were essential components at the beginning of life, how can they be today only parasites, unable to replicate autonomously? There is a tendency in evolution to increase genome sizes. However, the opposite exists as well and is often less obvious. Nature also reduces genes or gets rid of them to reduce the burden and energy requirements for their maintenance and replication. Viruses gain or lose genes to adapt to their environments. One experiment demonstrating the loss of genes was performed by Sol Spiegelman in 1965, who incubated the RNA of the phage Q beta, 4217 nucleotides long, in a test tube in the presence of an enzyme, an RNA replicase, in a rich milieu. Serial transfers of aliquots and regrowth in a new test tube were performed. The RNA replicated faster and faster and became smaller and smaller, ending up as non-coding RNA of 218 nucleotides after 74 generations [70,71,72]. This short non-coding RNA has been designated as “Spiegelman’s Monster”. This study explains the success of viruses to become the most abundant entities on our planet, where up to 10^33^ particles exist [73].

This experiment is a model of how an autonomous biological entity can become dependent, how the ancestors of viruses and viroids may have been autonomous initially, but in a rich milieu they became parasitic. In line with these results, one may speculate that some of the viroids may have lost their intrinsic enzymatic activity and used cellular enzymes for replication. The rich cellular milieu may have allowed such a reduction of genes, which is also an energy-saving simplification. Typical viruses—today—do not have enough genes to live on their own. This parasitic lifestyle may have been adopted during evolution as a consequence of environmental luxury. Only today, viruses must rely on the machinery of the host, such as bacteria, human cells, or other organisms. One could envisage two arms of evolution, one with viruses increasing in numbers and not in size, and another one starting with viruses which by contrast evolved into larger entities, perhaps into giant viruses, and finally into subcellular structures and then into cells. That leads to today’s situation. Here, comes support from giant viruses.

## 6. Giant Viruses

Giant viruses have only recently been discovered during the last decades. They are larger in diameter and genome size than many bacteria, often comprising more than 1000 genes and more than 1.5 million basepairs [12,13]. They mimic bacteria in size and composition, hence the name ‘Mimivirus’ for the first discovered giant virus.

Giant viruses harbor large genomes that encode components required for protein synthesis such as genes for ribosomal proteins, tRNAs, transferases, and translation factors. Yet, they do not have the complete set required for independence from cells. Protein synthesis is the definition of life in classical textbooks. This puts the giant viruses in the middle between living and inanimate matter and indicates that the transition between the two extremes is continuous. They may have been autonomously replicating at some stage of development. Are they “unfinished” bacteria on their evolutionary path to real bacteria, or regressed bacteria that lost genes during evolution? They can host viruses themselves, designated as virophages, because of their similarity to bacteriophages or phages that infect bacteria [74]. This has not been a property of previously known viruses. Last year, researchers discovered that giant viruses contain genes they do not seem to need, namely stretches of DNA important for cellular, not viral, metabolism [75]. Recently a giant virus was shown to integrate into an algae genome, thus increasing its genetic make-up by giant viral genes ranging in size from 78 to 1782 genes. Some algae had an entire giant virus genome integrated, two algae even had the whole genomes of two giant viruses in their DNA, leading to about 10% of the algae’s total gene counts. Some of these giant viruses may have been around in the algae for a long time, millions of years. Some viral DNA had even acquired non-coding DNA, introns—which are normally never found in viruses. The new DNA can even originate from other algae—indicating a gene transfer—can allow the algae to improve their survival and may help to increase diversity. Such interactions have been going on since the origins of life, and they will continue [76].

Based on the endosymbiotic theory, or symbiogenesis, it is considered here that viruses may have undergone a reductive evolution. Viruses are parasites in today’s world, but they may have become intracellular parasites as a consequence of selection due to certain environmental conditions and energy-saving measures. In the prebiotic world, pre-cells or pre-viruses may have looked alike. They both may have evolved further. Thereby, the viruses starting from initially autonomous viroids or viroid-like RNAs and other viral species such as tRNA-like plant viruses or giant viruses, perhaps first became more complex and then evolved to host dependent parasitic viruses.

## 7. Endosymbiosis

Non-coding RNA from coding RNA can be produced today experimentally depending on the conditions, as exemplified by Spiegelman’s Monster described above. A rich milieu, in this case the presence of a replicating enzyme, ribonucleotides, etc., can lead to loss of genetic information and result even in loss of autonomy. Examples found in nature are the endosymbionts of bacteria, which evolved to mitochondria or chloroplasts [77,78]. They were originally independently replicating cells that became dependent on their host. The prokaryotes gave up autonomous replication to become intracellular life forms by eliminating or delegating 2700 genes to the nucleus and keeping about 37 genes to themselves, as described for chloroplasts. They specialized and developed into the powerhouses of the cell [79]. *Rickettsia* species provide another example of loss of independence. They exert an obligatory intracellular lifestyle, yet, they have evolved from extracellular bacteria. The intracellular forms lost more than 20% of their genes [80]. Another example of endosymbiosis may be the Poxviruses, which show a relationship with the eukaryotic nucleus and share fundamental features, such as the DNA polymerase and mRNA capping [81]. Poxviruses appear to be relatives of the mimiviruses or giant viruses mentioned above. They all replicate in the cytoplasms in viral “factories”. A virus supplying the cellular nucleus can also be considered as an extremely rare event during the evolution of life on Earth. One may speculate that, on the contrary, related events happen even frequently today by infectious agents. Viruses infecting host cells can be considered as analogous situations. Especially retroviruses can integrate their DNA copies into the host DNA genomes, even for many generations; they become endogenized. The human genome consists of about 50% of endogenous retroviral-like elements or the related retroelements [82]. Endosymbiosis, or symbiogenesis, can extend beyond “master and slave” relationships as for many endosymbionts, but may include mutual benefits and partnerships [83]. Endogenization has now been shown for giant viruses as well [76].

It is a hypothesis that the ancestors of viruses including viroids exerted an independent lifestyle and then lost their autonomy during evolution to become the host-dependent parasites as we know them today [11,16]. These pre-cellular ancestors of present-day viruses and viroids must have been capable of replicating and evolving, and thus fit the NASA definition of life as “self-sustaining chemical systems capable of Darwinian evolution”. Loss of genes in response to the environment is a general process and may be the explanation for parasites as evolutionary successors of previously autonomously replicating entities [84,85,86,87,88].

## 8. Conclusions

The message we give here is that the beginning of life must have been simple, and that viroid-like RNAs are good candidates for the first life-like biomolecules. Viroids are the smallest known entities that are able to replicate and undergo Darwinian evolution, one of the definitions of life. Today’s viroids require host cells for their replication, but their ancestors may have been capable of cell-independent replication and evolution before the emergence of cellular life. Their small size, their versatility, mutagenicity, and autonomy, solely based on structural information in the absence of the genetic code and protein synthesis, are features needed for other forms of life, as exemplified by the ribosomes whose catalytic activity depends on two ribozymes. One of them performs peptide synthesis, the other one, RNase P, processes the ribosomal RNA and trims the tRNA to its final structure. It is the essential transporter of amino acids as required by the genetic code of the mRNA. Consequently, we propose that viroids may serve as models for early forms of life on Earth. Analogous structural elements with catalytic functions required for replication may be present on other planets and celestial bodies as well as exoplanets. Terrestrial life-like entities with principle properties of the simple ribozymes or viroids could be envisaged to have evolved elsewhere. Thus, viroids may be considered as lead candidates or models in the search for analogous entities outside of the Earth in the universe.

## Figures and Tables

**Figure 1 ijms-22-03476-f001:**
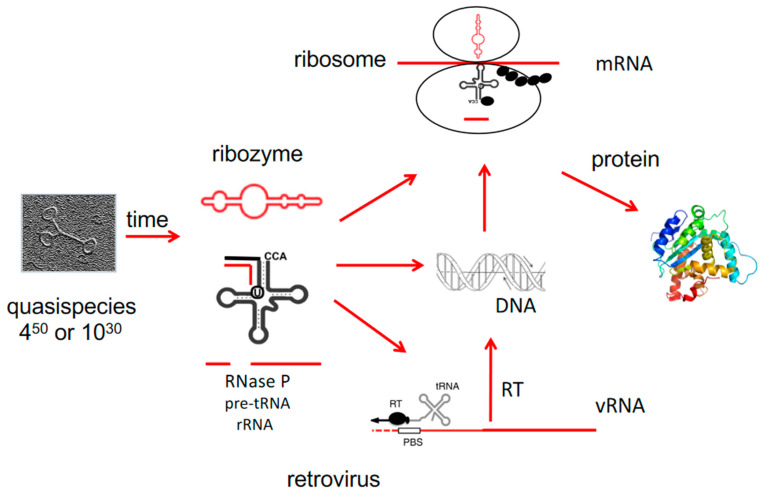
Viroids/viroid-like RNAs/ribozymes early during evolution. The picture shows a viroid. Viroid/ribozymes and tRNAs can form spontaneously from a quasispecies of many RNA molecules, both exhibit enzymatic activities. The ribozyme RNase P processes pre-tRNA to the shorter tRNA, DNA can arise from RNA without a reverse transcriptase (RT), protein synthesis in ribosomes requires ribozymes as essential components (mRNA is messenger RNA, vRNA is viral RNA, PBS is the primer binding site for tRNA binding) (For details see text).

**Figure 2 ijms-22-03476-f002:**
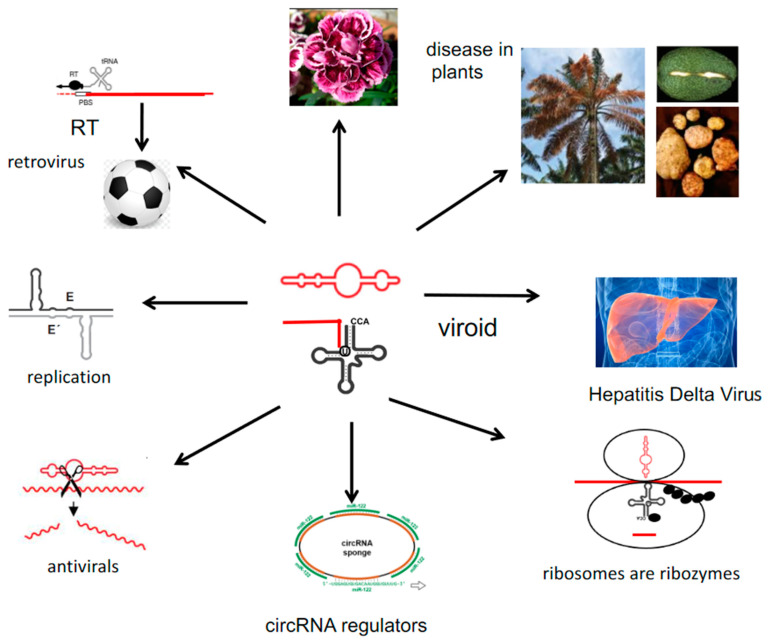
Multifunctionality of viroid-like RNA. Viroids/ribozymes and the ribozyme RNase P shown in the center, contribute to many phenomena, to diseases in plants such as palm trees, potatoes and avocado, to the human hepatitis delta virus, to protein synthesis in ribosomes, to regulatory circular RNAs, a chief-regulator, which regulates the levels of micro RNAs (miR), and to hammerhead ribozymes, symbolized as scissors, for gene therapy against cancer. Ribozymes can replicate in the test tube as shown in the model. The tRNA may be as ancient as the viroid and contributes to retrovirus replication. RT is the Reverse Transcriptase, which requires protein synthesis. The football symbolizes a retrovirus icosahedron. Carnations owe their pattern to a viroid and a para-retrovirus from plants (For details, see text) (RNA in red color).

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
