# Peer review of "Viroids and the Origin of Life"

_ijms, 2021, doi:10.3390/ijms22073476_

Round 1
Reviewer 1 Report
The manuscript promotes the hypothesis of RNA as the molecule at the origin of life and shows that viroids and viroid-like RNAs possess several (but not all) features required for "living fossils". The manuscript contains a large amount of quite convincing information and theoretical considerations.
The manuscript contains a few imprecise and/or misleading statements (listed below); a revision of these should not raise any problem.
In the manuscript two figures are mentioned; the "supplementary" Word file contains legends of these figures. But the figures are missing in the submitted files.
Abstract, Line 8:
"Viroids are non-coding circular RNA molecules with hairpin loop structures."
Any RNA secondary structure has to contain at least one hairpin; a circular RNA has to contain at least two hairpins. Thus I would use a different wording to emphasize the special structural features of viroids.
Line 9: "They [Viroids] are closely related to tibozymes ..."
Members of the family Avsunviroidae are ribozymes; they use this function to cleave, in cis, oligomeric replication intermediates to monomers. Thus what is meant by "closely related"?
Line 28: "The RNA of viroids and ribozymes is non-coding; information is based on charge and structure."
1. What is meant by "charge"? Because viroids and ribozymes consist of RNA, their charge is identical to that of any other RNA of equal size.
2. A few exceptional ribozymes are coding; one example is the RNA of HDV. Note that the same feature is given more carefully in line 63: "... viroids, which are composed of RNA and typically do not encode any proteins."
Line 35: "In-ice evolution of ribozymes has been proposed as another option for the origin of life (Lehman et al., 2013)." I don't find a mention on "in-ice" evolution in this reference.
Lines 253f: "Known from history is Cadang-Cadang a disease of trees caused by a viroid in the Philippines affecting coconut trees, where the virus was transmitted by the natives using a single knife to cut off the fruits of many trees (Randles et al., 1988). This caused transmission of the viroid and destroyed whole forests."
1. To my knowledge, the disease is not only of historical importance but still ongoing in Indonesia, Malaysia, West Africa, Central America, and further regions.
2. I would prefer the word "workers" (or similar) instead of "natives".
3. Transmission by cutting is not mentioned in the given reference.
4. To my knowledge the disease mainly affected commercial plantations but not forests.
Ref. 29: "protein-pnly" => "protein-only"
Ref. 65: "perspective Mol." => "perspective. Mol."
Ref. 58: "Moniruzzam M.; " => "Moniruzzaman M.; "
Author Response
see uploaded attachment

Reviewer 2 Report
The review “Viroids and the Origin of Life” offers an interesting perspective on the possible connection existing between viroids and the first building blocks at the origin of life and other primordial entities such as viruses and ribozymes.
The information presented is not always up-to-date and sometimes a bit difficult to follow.
To improve the understanding of some of the implications and put information in to context, a distinct section, in the first part of the review, containing some basic information about viroids structure and functions would be of great help. Some information is already present in the text from line 244 to 282. In particular, what are the catalytic abilities of the viroids characterized so far? Which viroids contain the hammerhead ribozyme? What is the mechanism, if known, by which the presence of a viroid, which does not generate proteins, causes a disease and what information about their function/origin can be inferred?
Oftentimes the terms viroids and ribozyme are used almost interchangeably and this creates some ambiguity, e.g. from line 101 to 106. Are all viroids ribozymes? Is this paragraph describing viroids or ribozymes? Please try to be more specific throughout the text when referring to viroids in particular or to ribozymes more in general.
Line 38 and paragraph at line 168: the Miller-Uray experiment demonstrated the production of amino acids only, not RNA precursors. However, some later experiment were able to generate nucleobases. Please include the appropriate citations.
Line 156: What relevant biomolecules were generated?
Line 51: which more complex viruses?
At line 139, the reference Goodman et al. 2012 does not mention retroviruses and it should be moved to line 137.
At line 246-248, please add a reference for the binding of regulatory factors to non-coding regions.
Paragraph at line 283: There are actually many circRNAs with miRNA-sponge ability that have been identified in animals so far. A huge number of circRNAs have been identified in plants as well, although it is still not clear whether the miRNA sponge ability is their major function. Is there experimental evidence - or lack thereof, of a similar mechanism for viroids in plants?
Fig.1 legend. Do you imply that circRNAs are viroids? Perhaps it would be better to change the title of the figure to “Multifunctionality of viroid-like RNA”.
Author Response
uploaded as attachment

Round 2
Reviewer 2 Report
The manuscript greatly improved and can be accepted in the present form.
Author Response
1. done
2.accepted and quoted as suggested, was wrong before thanks!
3.done
4.papaya was wron chaned to avocado in the legend and new Fig 1
5. deleted

This manuscript is a resubmission of an earlier submission. The following is a list of the peer review reports and author responses from that submission.
Round 1
Reviewer 1 Report
Line 2: Throughout the article it is noted that not „viroids“ but „viroid like RNAs“ are at the origin of life. Therefore I would suggest to change title into: „Viroid-like RNAs and the Origin of Life“
Line 12/13: In the abstract there is the sentence „Furthermore, they are the essential elements for protein synthesis of cellular organisms as parts of ribosomes“. This is not correct. Viroids are currently considered infecting agents of plants. The sentence is correct only if it is clearlified that this are „viroid-like RNAs“.
Line 23-26: Please add reference:
Briones C, Stich M, Manrubia SC. The dawn of the RNA World: toward functional complexity through ligation of random RNA oligomers. RNA. 2009 May;15(5):743-9. doi: 10.1261/rna.1488609.
Line 27: Add to „In theory, any chemical reaction can be carried out by ribozymes“, „In theory, any chemical reaction within biotic agents can be carried out by ribozymes“.
Line 29-34: The thesis presented here is one favorite. A contrasting favorite should be mentioned also. Therefore add:
Lehman N. Origin of life: Cold-hearted RNA heats up life. Nat Chem. 2013 Dec;5(12):987-9. doi: 10.1038/nchem.1811.
Line 53: There are other considerations also (virocell concept). Therefore add:
Forterre P, Prangishvili D. The major role of viruses in cellular evolution: facts and hypotheses. Curr Opin Virol. 2013 Oct;3(5):558-65. doi: 10.1016/j.coviro.2013.06.013.
Line 54-58 „origin of life“. But what is Life? Add references here:
Schroedinger, E. (1944). What is Life? The Physical Aspect of the Living Cell, 1944, London: Cambridge University Press.
Update: Witzany, G. (2020) What is Life? Front. Astron. Space Sci. 7:7. doi: 10.3389/fspas.2020.00007
Line 64-71: definition of NASA „Life is a self-sustaining chemical system...“ depends on systems theoretical perspective which can be applied to living cells also. But there are also other theoretical approaches outside from systems theory, because first of all a cell is a cell not a system.
Also the next sentence „machines as hardware and programs as software. RNA is the only biological molecule we know of that is both "software and hardware" in one. RNA can be a self-replicating chemical entity, and at the same time also a carrier of information.“ is rather metaphorical. A machine is an mechanistically functioning human artefact. Are cells really mechanistic artefacts? Additionally, computer software depends on algorithm based programming. But the genetic content of cells cannot be understood by algorithm based mechanisms, because under the superficial nucleotide grammar there are many possibilities to epigenetically mark the sequence for different proteins, which means there is a deep grammar also, which cannot be identified by algorithm based investigations of nucleotide sequences. Additionally the term „Information“ one can find 60 different definitions. Which one is used here? Suggestion: delete paragraph line 64-71
Lines 72: delete „Nobel ,prize laureate“. Empirical data are not evaluated by awards but by verification/falsification
Line 75-77: the quasispecies concept is of main importance here. therefore it should – together with update – be referenced:
Eigen M. Selforganization of matter and the evolution of biological macromolecules. Naturwissenschaften. 1971 Oct;58(10):465-523. doi: 10.1007/BF00623322.
Villarreal LP, Witzany G. Rethinking quasispecies theory: From fittest type to cooperative consortia. World J Biol Chem. 2013 Nov 26;4(4):79-90. doi: 10.4331/wjbc.v4.i4.79.
Line 82: „... a primitive metabolism...“ change into „... a proto-metabolism“
Line 87-88 redundant: „The smallest replicating entities one can think of are the viroids or ribozymes, both of which can exhibit catalytic enzymatic functions.“
Line 97-98: „It has been suggested, that ribozymes can even replace the function of a replicase and perform RNA polymerization.“ Reference?
Line 106-107: „Life, however, likely started at hydrothermal vents deep in the ocean where sunlight cannot reach, yet, for example, geothermal energy is available.“
Add contrasting approaches also:
Attwater J, Wochner A, Holliger P. In-ice evolution of RNA polymerase ribozyme activity. Nat Chem. 2013 Dec;5(12):1011-8. doi: 10.1038/nchem.1781.
Line 165: „..RNAse P recognizes structures...“ Recognition is based on „cognition“. Therefore exchange this anthropomorphistic term into „identifies“
Line 178-179 Add reference:
Petrov AS, Bernier CR, Hsiao C, Norris AM, Kovacs NA, Waterbury CC, Stepanov VG, Harvey SC, Fox GE, Wartell RM, Hud NV, Williams LD. Evolution of the ribosome at atomic resolution. Proc Natl Acad Sci U S A. 2014 Jul 15;111(28):10251-6. doi: 10.1073/pnas.1407205111.
Caetano-Anollés G, Sun FJ. The natural history of transfer RNA and its interactions with the ribosome. Front Genet. 2014 May 9;5:127. doi: 10.3389/fgene.2014.00127.
Or: Caetano-Anollés D, Caetano-Anollés G. Commentary: History of the ribosome and the origin of translation. Front Mol Biosci. 2017 Jan 10;3:87. doi: 10.3389/fmolb.2016.00087.
Line 190-191: chapter 4 is too long. Maybe create a second headline here „“Transition from RNA to DNA“
Line 243-244: :“They are reminiscent of viroid RNAs“. Correct into „viroid-like RNAs“.
Line 256: „... evolution of cells.“ Virus first perspective is initiated by LP Villarreal. Add reference:
LP Villarreal. Viruses and the Evolution of Life, 2005, ASM Press, Washington.
Line 276: „Instead they must rely on the machinery of the host....“ A machine is a metaphor for mechnistically functioning human artefact. Cells in contrast are not human artefacts and do not function mechanistically
Chaper 7: Endysymbiosis: Please add some sentences on the origin of the eukaryotic nucleus and add references:
Takemura M. Medusavirus. Ancestor in a Proto-Eukaryotic Cell: Updating the Hypothesis for the Viral Origin of the Nucleus. Front. Microbiol.2020, 11:571831. doi: 10.3389/fmicb.2020.571831
Bell PJL. Evidence supporting a viral origin of the eukaryotic nucleus. Virus Res. 2020 Nov;289:198168. doi: 10.1016/j.virusres.2020.198168.
Chaikeeratisak V, Nguyen K, Khanna K, Brilot AF, Erb ML, Coker JK, Vavilina A, Newton GL, Buschauer R, Pogliano K, Villa E, Agard DA, Pogliano J. Assembly of a nucleus-like structure during viral replication in bacteria. Science. 2017 Jan 13;355(6321):194-197. doi: 10.1126/science.aal2130.
Author Response
Answers to Reviewer number 1
Answers by K Moelling
Al Queries have been answered and fulfilled .Except the following.
2 and title
It clearly states in the abstract viroid or viroid-like. I do not want to change the whole manuscript based on the reviewers opinion. I really mean viroids.
12/13 I clarified this, it refers to ribozymes.
23-26 inserted
27 accepted
39-34 ref added
53 ref included and a sentence also
54-58 This is a discussion about a simple biomolecule, the viroids, both references do not fit,
my explanation is in line 64-71, NASA and F. Dyson.
64-71 This is the reviewer`s opinion and a suggestion, no request for changes
I do not want to delete that paragraph. It refers to Freeman Dyson and I want to leave this in. the reviewer suggests a deletion, but I really want to refer to f Dyson and want to leave the paragraph in.
72 Szostak is a Nobel prize winner - why delete that?
75-77 both references are inserted
82 is a matter of taste or a new definition
87-88 the sentence is ok
97-98 ref was given at end of next sentence line 100.
106 ref included now
165 is a matter of taste.
178-9 ref Petrov included
190 new subtitle does not fit,
243 my sentence is correct
256 inserted new ref
276 is a matter of definition or taste.
Chapter 7 endosymbiosis , i agree, new sentences on Pox viruses inserted as proposed by the reviewer and Ref Bell 2020 included.
New ref are included , total of 9 references as suggested by the reviewer.
Briones, Lehmann, Forterre, Eigen, Villarreal. Altwater, Petrov, Villarreal 2005, Bell
Reviewer 2 Report
This manuscript covers historical background and is very interesting to read as a story. However, there are many statements based on conjecture or belief (e.g., lines 57-58, 181-182, 239-240), and I don't think it is appropriate to publish it as a review. It would be less problematic to discuss the possibility of past events, which cannot be confirmed, without being definitive. Nevertheless, this manuscript may be better dealt as a hypothesis than a review.
As for what needs to be fixed, I would like to make the following comments.
Some important references are missing (for the sentences in lines 120-122, 137-139, 222-224, 233-235, 295-296, 296-298, 298-300, 301-302).
The intent of the statement in lines 137-140 is unclear.
The expression "...as pointed out by Szostak in a university lecture that can be viewed on Youtube... (lines 206-207)" is not appropriate as a basis for a scientific paper.
Some of the ' have become ` (Line 32 and 299).
I cannot find Fig.1 (line 182).
Author Response
120 - 122 fullerene new reference added:
Goodman G, Gershwin ME, Bercovich D Fullerene and the origin of life.
.Isr Med Assoc J. 2012 Oct;14(10):602-6.
137-139 exoplanets
I added a sentence on exoplanets and the possibility of extraterrestrial life and a new ref.
Turnbull MC 2014 Finding planets and life among the stars. EMBO Rep. 2014 Oct;15(10):1002-9. doi: 10.15252/embr.201439314
222-224 cadang added new Ref:
Randles, JW, Rodrigues MJB Imperial JS (1988) Cadang-cadang disease of coconut palm Microbiol Science 5(1);18-22
233- 235 delta
Flores et al.,2014; Taylor 2009 both Ref were given in line 238 and 240
295-296 new giant virus , the ref was in line 304
because the whole paragraph refers to this reference, an additional ref is included.
Pennisi E. Giant virus genomes discovered lurking in DNA of common algae , Sciencemag 11,1-6 2020 doi:10.1126/science.abf7767
296 - 298 -300,-302 all the information is covered by the ref in line 306
137-140 unclear
I added a sentence for better understanding and a new reference (Turnbull, 2014).
32 and 299
I do not understand
206-207 Szostak ref:
iIeliminated the youtube reference and added a new one,
Walton, Travis; Szostak, Jack (2016). "A Highly Reactive Imidazolium-Bridged Dinucleotide Intermediate in Nonenzymatic RNA Primer Extension". Journal of the American Chemical Society. 138 (36): 11996–12002. doi:10.1021/jacs.6b07977.
Fig 1 needs to be added I will enclose it as pdf.